# Message framing materials applied to healthy eating decision-making for pregnant women with gestational diabetes mellitus: An exploratory study

Xia Han[1], Nuo Xu[2], Sijing Chen[3], Jingjing Zhang[3], Ping Gu[1]*

1 School of Nursing, Nanjing Medical University, Nanjing, China, 2 Department of Nursing, Changzhou Health Vocational Technology College, Changzhou, China, 3 Department of Maternity, Women's Hospital of Nanjing Medical University, Nanjing, China

* gpapple83@163.com

## Abstract

### Aims

To develop and initially validate message framing materials for promoting healthy dietary decisions in GDM populations.

### Methods

The Delphi survey involved 17 experts, and consensus was obtained after two rounds. In Pre-survey I, 30 participants randomly selected one material (gain framing vs. loss framing) and complete a manipulation check item to ensure that the stimulus material manipulation was valid. Pre-survey II involved 60 participants who rated all messages using a Likert 5 rating questionnaire, a Wilcoxon signed rank sum test was used to examine the message framing effects.

### Results

After two rounds of Delphi surveys, experts reached consensus on the final materials, which contained two message framings, each containing 11 entries. The manipulation test for materials achieved 100% validity. The findings indicated significant message framing effects on healthy eating decisions among GDM populations, with loss-framed messages proving more persuasive.

### Conclusion

Scientifically valid message framing materials were developed and initially applied, providing an empirical basis and direction for future related research.

**Data availability statement:** All relevant data are within the manuscript and its Supporting Information files.

**Funding:** This work was supported by the Project of 'Nursing Science' Funded by the 4th Priority Discipline Development Program of Jiangsu Higher Education Institutions (Jiangsu Education Department （2023）No.11) and the Priority Academic Program Development of Jiangsu Higher Education Institutions (2018, No. 87). The funders had no role in study design, data collection and analysis, decision to publish, or preparation of the manuscript.

**Competing interests:** The authors have declared that no competing interests exist.

**Abbreviations:** GDM, gestational diabetes mellitus; IDF, International Diabetes Federation; HAPO-FUS, Hyperglycemia and Adverse Pregnancy Outcome Follow-up Study; FIGO, International Federation of Gynecology and Obstetrics; ADA, American Diabetes Association; g, gram; kcal, kilocalories; fMRI, functional magnetic resonance imaging; EEG, Electroencephalogram; ERPs, Event Rated Potentials; RCTs, Randomized Controlled Trials; TTM, Trans-Theoretical Model; HBM, Health Belief Model.

## Introduction

Globally, gestational diabetes mellitus (GDM) prevalence has risen due to factors such as increasing maternal age and the adoption of the revised diagnostic criteria and procedures for GDM [1]. This growing prevalence represents a significant public health challenge, highlighting the importance of active prevention and treatment to improve the health of mothers and their children, as well as enhance the overall quality of life [2]. Nutritional therapy plays a crucial role in improving maternal and infant health outcomes. However, studies indicate that pregnant women with GDM often struggle with poor dietary self-management [3,4]. The American Diabetes Association (ADA) [5] stated that healthy dietary decision-making is the most challenging aspect of improving self-management of GDM. Difficulties in dietary decision-making and poor choices are significant barriers to effective self-management in diabetes patients [6], and a lack of relevant health education, combined with confusion regarding health information, may exacerbate this issue [7]. A major challenge in health education is that: Health information is mostly based on normative, rigorous, and objective medical terms, and there can be a big difference in how audiences understand and prefer information. Furthermore, the limited scope of available information often drives individuals to seek information from other sources, such as online sources [8]. Therefore, it is essential to seek an effective health information communication strategy that can motivate people to change unhealthy behaviors or adopt healthy behaviors.

## Background

The 10th edition of the Diabetes Atlas published by the International Diabetes Federation (IDF) [9] indicates that 16.7% of women worldwide experience varying degrees of hyperglycemia during pregnancy, with gestational diabetes mellitus accounting for 80.3% of these cases, making it the most common complication of pregnancy. The development of diabetes significantly increases maternal and fetal risk [10]. A study of 7,506,061 pregnant women from 156 studies, controlling for confounding factors such as maternal body mass index, revealed that pregnant women with GDM had higher odds of cesarean section, preterm delivery, macrosomia, low 1-minute Apgar scores, and older-than-gestational-age babies at birth, compared to pregnant women with normal gestational glucose levels [11]. Further, women with GDM face a significantly higher risk of developing diabetes and cardiometabolic disease, as well as their offspring [1,12].

The International Federation of Gynecology and Obstetrics (FIGO) has stated [13] that effective dietary management is crucial for improving maternal and fetal health in pregnant women with GDM. GDM can be managed through diet in 70–80% of cases [14]. Nevertheless, a study [4] involving 950 pregnant women with GDM found that 56.95% of patients had medium or low levels of dietary self-management, and Mustafa et al. [3] also found that only 104 (33.2%) of 313 pregnant women with GDM adhered well to nutritional advice. In response to this dilemma, research has shown that receiving more relevant health information can enhance adherence to dietary recommendations and promote healthier behaviors [15,16]. However, a key challenge remains: how to effectively increase individuals' acceptance and recognition of health information, which is essential for overcoming these barriers.

The message framing effect, first proposed in 1997, introduced new perspectives and methods for understanding behavioral decision-making and influencing behaviors across various domains [17]. Message framings are categorized according to how the message is presented: gain framing (which emphasizes the benefits of engagement as recommended) and loss framing (which emphasizes the harms of not engaging as recommended) [18]. The message framing effect suggests that health information does not always positively influence individuals'

health behaviors. It posits that different types of message framings can lead to variations in decision-making, and puts forward the main idea that decision-makers are "limited rational" individuals, challenging the traditional notion of consistency in rational decision-making [18]. The Elaboration Likelihood Model (ELM) also affirms this viewpoint, arguing that there are two different ways of individual information processing: central path processing and edge path processing [19]. In general, individuals tend to rely on edge path processing to make limited rational decisions [20].

The message framing effect is an effective strategy for nudging people's health behaviors. It has been shown to offer significant benefits for pregnant women and people with diabetes. Studies found that framed messages were more likely to promote oral health in pregnant women [21]. Gain-framed messages have proven more effective in encouraging smoking cessation among female smokers of childbearing age [22]. And in addition, patients with type 2 diabetes have demonstrated improved their self-management behaviors and quality of life after receiving framed messages [23]. However, no studies have specifically examined the impact of the message framing effect on pregnant women with GDM, and extrapolating previous studies on pregnant women and diabetes patients to this population is not straightforward. More studies are needed to explore the potential impact of message framing effect on pregnant women with GDM.

Given that message framing effect is a promising strategy for promoting healthy eating behaviors in women with GDM, this study developed GDM-specific healthy eating message framing materials for initial application and validation. To lay the groundwork for future research that leverages message framing effects to enhance the acceptance of healthy eating information among pregnant women with GDM. This approach aims to facilitate exploration of the message framing effects in this area and provide new strategies and tools for the effective dissemination of healthy eating education in the future.

## Methods

### Aim

To develop message framing materials for healthy eating for promoting healthy eating among pregnant women with GDM, and to conduct preliminary application and validation. This would provide practical, effective stimulus materials and an empirical evidence base for future research on the application of message framing effects to promote healthy eating in GDM.

### Research design and methods

This study was divided into three phases, see Fig 1 for specific research procedures:

- Phase 1: Preparation for Delphi

Through literature review and discussion among the research team, a pool of items for the healthy diet message framing materials was compiled.

- Phase 2: Delphi Survey

Further modifications and refinements were made to the raw materials using Delphi.

- Phase 3: Pre-surveys

Pre-surveying the study materials to ensure that confusing statements in the stimulus materials were clear and the independent variables were easily identifiable; Exploring and validating message framing effects in dietary information decision-making in pregnant women with GDM in a preliminary study.

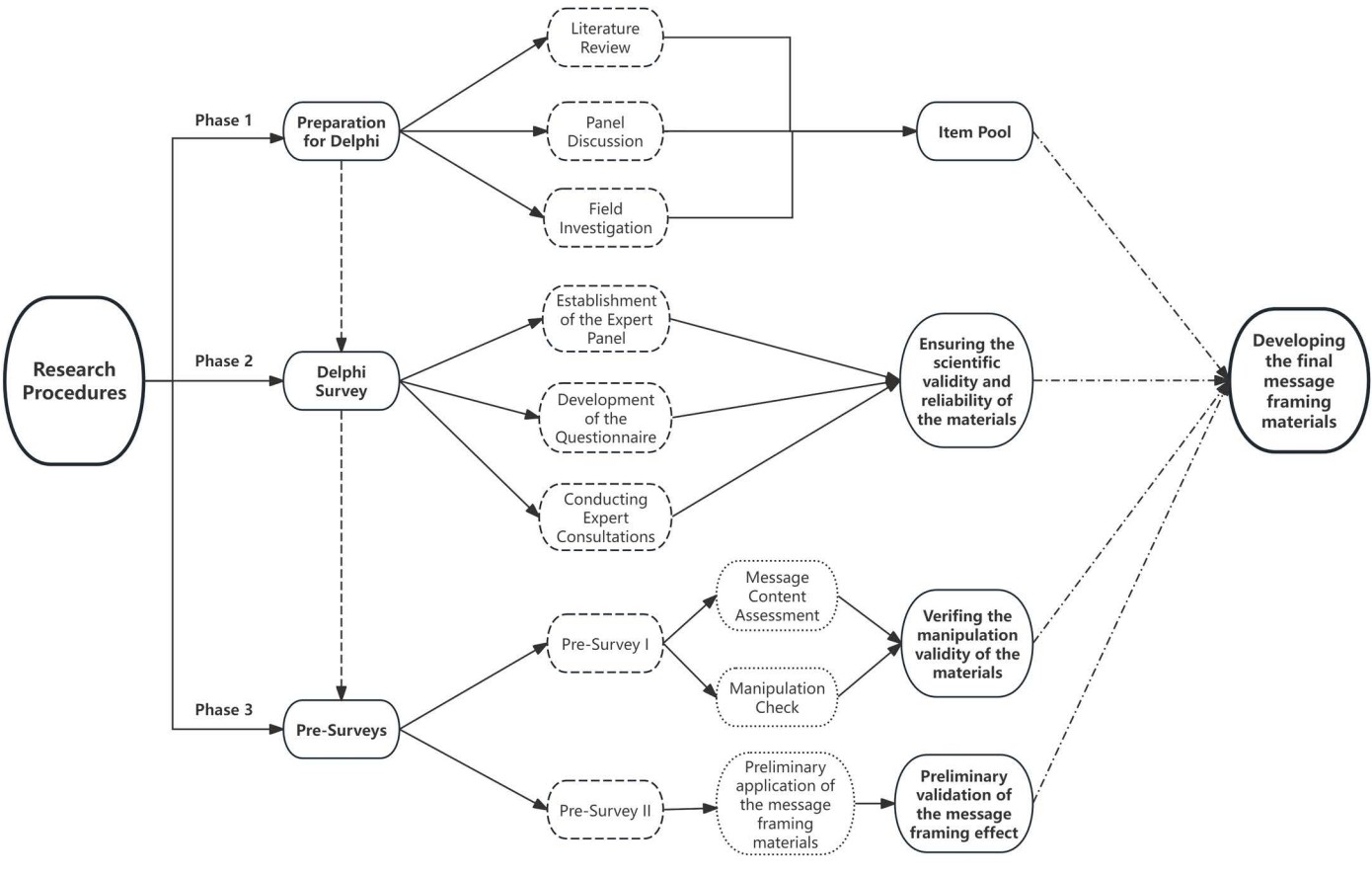

**Fig 1. Research procedures.**

## Preparation for Delphi

Before initiating the consensus process, we compiled a pool of items for the healthy diet message framing materials through a literature review and research group discussions.

The American Diabetes Association (ADA), Up to Date, PubMed, CNKI, Wanfang database, CBM, and Medlive databases were searched using the key terms "diabetes*, gestation/gestational diabetes mellitus/GDM" and "diet*/dietary advice/dietary behaviors/nutrition/nutritional therapy" to obtain healthy diet information related to gestational diabetes. Based on Gao Ruitong's study [23], combined with the results of the research group discussion and this review, we chose two dimensions, i.e., immediate/long-term effects on mothers and infants, with staple foods, meat, oils, fruits, vegetables, cooking styles, and eating habits as the main themes for dietary information selection, and framed the information to form a pool of items of materials for this study.

The validity of providing information that focuses on the health consequences of food choices has already been established, particularly when messages are formulated in prefactual terms (i.e., "if... then" plans) [24]. In this study, messages were framed using "if... (then)" scenarios, highlighting both positive (gain) and negative (loss) outcomes, taking "cooking styles" for example:

- Gain framing: "If you often choose to steam, stir-fry, or boil your cooking, it may be beneficial to control your blood sugar and weight."

- Loss framing: "If you often choose to fry, thicken, or deep-fry your cooking, it may lead to fluctuations in your blood sugar and weight gain."

The materials were designed with a combination of text and graphics, and variations in word count and frequency were minimized to control for confounding variables and ensure participants had similar levels of comprehension.

## Delphi survey

**Design and method.** For this study, a two-round Delphi survey was performed and in accordance with Conducting and REporting of DElphi Studies (CREDES) checklist [25]. The Delphi method, which was created in 1950 by Olaf Helmer at the Rand Corporation [26], aims to converge the opinions of a group of experts in order to reach consensus on particular topics through rounds of questionnaires [27]. This method involves a structured communication process with a defined panel of experts and is based on the principle of individual and anonymous exchanges. The purpose of this study was to identify and refine the most pertinent items for our materials [28].

**Establishment of the expert panel.**
**Inclusion criteria:**

1. A minimum of 5 years of relevant professional experience;

2. At least a bachelor's degree;

3. An intermediate-level or higher professional title;

4. Informed consent and active participation in the study.

**Exclusion criteria:**

1. Individuals who declined to participate in the study;

2. Experts who did not participate in all rounds of the Delphi survey

Humphrey-M et al. [29] proposed that the number of experts required for the Delphi method is 5–20. For this study, 19 experts were initially selected. However, after excluding the two experts who did not respond within the time limit in the second round, a total of 17 experts were finalized for the Delphi survey.

**Development of the questionnaire.** The following information is contained in the consultation letter that the research group jointly designed:

1. **Study Introduction:** Background, purpose, and content of this study.

2. **Expert Information:**

1) **Basic information of experts**: Name, gender, age, education, title, position, length of service, unit, etc.;

2) **Judgment of the content**: Questionnaire mainly from the theoretical analysis, practical experience, understanding of domestic and foreign counterparts, intuition in four aspects of the evaluation, and each aspect of the three grades;

3) **Familiarity with the content**: According to the expert's familiarity, it is divided into five grades: very familiar, familiar, general, not too familiar, and unfamiliar.

3. **Importance rating and opinion:** Using the Likert 5-level scoring method to rate the importance of the items (1 = very unimportant, 5 = very important) [30]. Meanwhile, the table is provided with the column of modification opinions and increase/decrease opinions.

**Delphi procedures.** From September to November 2023, two rounds of expert consultation were conducted, with an interval of one month between each round. This study was approved by the Medical Ethics Committee of Women's Hospital of Nanjing Medical University (2022KY-114). Prior to the survey, the basic requirements and precautions were explained to the experts. Written or Electronic Informed Consent was obtained from all experts, and the expert consultation questionnaire was distributed either via email or in person. Experts were requested to provide their responses within two weeks.

**Statistical analysis.** SPSS 25.0 software was used to statistically analyze the data. Count information is expressed as frequency (percentage), and measurement information is expressed as ($\bar{x} \pm s$). The screening criteria for the items of this study were: the mean of importance score ($\bar{x}$) > 3.5 and the coefficient of variation (CV) < 0.25 [31]. The expert authority coefficient (Cr) reflects the level of expert knowledge of the research problem. The formula for calculating it is: Cr = (Ca + Cs)/2, where Ca stands for the judgment coefficient of experts and Cs stands for the familiarity coefficient of experts [32]. Usually, the Kendall harmonization coefficient (W) is used to indicate the reliability of raters. W value ranges from 0 to 1 [32].

## Pre-surveys

**Inclusion and exclusion criteria.**
**Inclusion criteria:**

1. Initial diagnosis of GDM during pregnancy and met the diagnostic criteria for Guideline of diagnosis and treatment of hyperglycemia in pregnancy (2022) [33];

2. Age ≥ 18 years;

3. Those who have a basic comprehension of reading and no communication disabilities.

**Exclusion criteria:**

1. Pregnant women with GDM who had pregestational diabetes mellitus, multiple pregnancies, and combined severe medical, surgical, or obstetric complications were excluded.

**Sample size.**

**Pre-survey I:** Based on the sample size calculation method used in previous studies [34], a sample of 30 participants was determined for this pre-survey.

**Pre-survey II:** G＊Power 3.1 software was used to calculate the required sample size. With a medium effect size ($d = 0.50$), significance level ($\alpha = 0.05$), and statistical power of $1\text{-}\beta = 0.90$, a minimum of 47 subjects was determined [35]. To account for potential sample ineligibility, the sample size was increased by 15%, resulting in a final required sample size of at least 55 participants. Ultimately, 60 participants took part in this study.

**Methods and data collection procedures.** This study recruited pregnant women with gestational diabetes mellitus who came to hospital in Nanjing, China, between February 20 and March 24, 2024, and met the above inclusion criteria. This study was approved by the Medical Ethics Committee of Women's Hospital of Nanjing Medical University (2022KY-114). The significance, objectives and survey requirements were explained to all participants prior to the conduct of the study. Following the receipt of written informed consent, the formal data collection process was initiated.

**Pre-survey I:** Using a purposive sampling method, 30 participants were randomly assigned to one of two questionnaires (gain framing or loss framing). Participants completed the questionnaires based on their actual situation.

**Pre-survey II:** Using a purposive sampling method, an additional 60 participants recruited and exposed to both message framings. The purpose and requirements were explained, and participants completed the questionnaires on-site.

**Measures.**
**Pre-survey I**

1. Message Content Validation: Two questions were developed to assess participants' attention to the content: a fill-in-the-blank question and a multiple-choice question, aimed at identifying those who did not carefully read the material;

2. Manipulation Check: One question, set to "What is the main story of the above material? " And provided two choices of answers: "The benefits of a healthy diet for mothers and their babies" and "The harms of an unhealthy diet for mothers and their babies" to test the manipulation effect of the message framings;

3. Participant Information: Basic information about the participants.

**Pre-survey II**

1. Messages Agreement: Using a random number table to sort all messages containing both message framings, participants were asked to measure their agreement with each message using a Likert 5 rating questionnaire (1 = completely disagree, 5 = completely agree). To gain a preliminary understanding of the participants' preference for the two framings of messages by comparing the scores obtained from the different message framings of a homogenous healthy eating message;

2. Participant Information: Basic information about the participants.

**Statistical analysis.** Microsoft Excel 2021 was used for data entry and SPSS 25.0 was used for statistical analysis. Count information is expressed as frequency (percentage), and measurement information is expressed as ($\bar{x}$ ±s). A Wilcoxon signed rank sum test was used to examine the message framing effects in healthy eating information decision-making among pregnant women with GDM. This nonparametric method was chosen because the data in the Pre-Survey II section did not meet the assumption of normality, making parametric tests inappropriate. The Wilcoxon signed-rank sum test is ideal for comparing differences between two related (paired) samples, especially when the data cannot be assumed to follow a normal distribution [36]. Differences were considered statistically significant at P < 0.05.

## Results

### Results of the Delphi survey

**Characteristics of Delphi experts.** Table 1 describes the characteristics of Delphi consulting experts. Most of the experts are 30 to 49 years old, with an average working year of 23.1 years. The panel has a higher proportion of female experts. Their professional titles are intermediate or above, and their educational qualifications are undergraduate or above. The professional fields of experts involve Obstetrics and gynecology, Nutriology, Endocrinology, Psychology. Therefore, the expert panel is characterized by extensive experience, high professional qualifications, and a broad range of disciplines, ensuring the comprehensiveness and reliability of the research findings [37].

**Delphi survey results.** The questionnaire recall rates for both rounds were over 85% (Table 2), indicating that the experts were highly motivated. The expert authority coefficients were 0.905 and 0.923 (Table 3), indicating that the results of the Delphi have a high degree of authority. Table 4 shows the Kendall's harmony coefficient (W), which was 0.171

**Table 1. Characteristics of Delphi experts.**

| Characteristics | | Number (N = 17) | Percentage (%) |
|---|---|---|---|
| Age | 30–39 | 8 | 47.06 |
| | 40–49 | 8 | 47.06 |
| | ≥50 | 1 | 5.88 |
| Gender | Male | 5 | 29.41 |
| | Female | 12 | 70.59 |
| Professional title | Intermediate | 7 | 41.18 |
| | Sub-senior | 5 | 29.41 |
| | Senior | 5 | 29.41 |
| Years of work | <10 | 2 | 11.76 |
| | 10–19 | 9 | 58.82 |
| | >20 | 6 | 35.29 |
| Employers | Hospital | 16 | 94.12 |
| | University | 1 | 5.88 |
| Fields of research | Obstetrics and gynecology | 6 | 35.29 |
| | Nutriology | 6 | 35.29 |
| | Endocrinology | 3 | 17.65 |
| | Psychology | 2 | 11.76 |

**Table 2. Experts' response rate of two rounds of Delphi.**

| | 1st round | 2nd round |
|---|---|---|
| Questionnaire delivered | 19 | 19 |
| Questionnaire received | 19 | 17 |
| Valid questionnaire | 19 | 17 |
| Rate of response (%) | 100.00% | 89.47% |
| Rate of valid questionnaire (%) | 100.00% | 100.00% |

($\chi2 = 61.142$) in the first round and 0.258 ($\chi2 = 91.951$) in the second round, demonstrating consistent coordination and supporting the reliability of the findings. The average scores of questionnaire dimensions and item importance ($\bar{x}$) is more than 3.5, and the coefficient of variation (CV) is less than 0.25, indicating a high degree of consensus among the experts. Therefore, no items were deleted [31].

After the first Delphi round, we received nine comments and suggestions, such as "The 'Ketosis' terminology is not well understood by patients; does it need to be changed?" and "Shouldn't there be an example of what a 'non-fermented and chunky food' is?" Experts recommended revising terminology and clarifying certain concepts. After consulting the literature and discussion, the research team made adjustments for the second round; the second round of Delphi expert opinions tended to converge, and no further changes to the entries, finalizing the materials. The content and scoring of the material entries from both Delphi rounds are shown in the animation (S1 Appendix).

## Results of the pre-survey I

**Characteristics of the participants.** We recruited a total of 30 pregnant women with GDM. There were 15 participants in each group, and the information about the characteristics of the participants is shown in Table 5.

**Table 3. Levels of authority of the Delphi experts.**

| Round | Ca | Cs | *Cr |
|---|---|---|---|
| 1st round | 0.959 | 0.850 | 0.905 |
| 2nd round | 0.965 | 0.880 | 0.923 |
| Average | 0.962 | 0.865 | 0.914 |

*Cr = (Ca + Cs)/2

**Table 4. Expert coordination degree and coefficient of variation.**

| Round | Kendall W | $\chi^2$ | P | CV |
|---|---|---|---|---|
| 1st round | 0.171 | 61.142 | 0.000* | 0.14 |
| 2nd round | 0.258 | 91.951 | 0.004* | 0.09 |

*$P < 0.05$.

**Table 5. Characteristics of the study participants (n = 30).**

| Variables | Characteristics | Gain framing (n = 15) | Loss framing (n = 15) |
|---|---|---|---|
| Age (years) | | 33.47 ± 4.55 | 32.80 ± 4.25 |
| Pre-pregnancy BMI (kg/m²) | | 22.14 ± 3.16 | 22.48 ± 3.35 |
| Ethnicity | Han nationality | 15 (100) | 15 (100) |
| | Minority | 0 (0) | 0 (0) |
| Educational level | Primary school and below | 0 (0) | 0 (0) |
| | Junior high school education | 0 (0) | 0 (0) |
| | High school or post-secondary education | 0 (0) | 1 (6.7) |
| | College degree | 4 (26.7) | 5 (33.3) |
| | Bachelor's degree | 8 (53.3) | 5 (33.3) |
| | Master's degree and above | 3 (20.0) | 4 (26.7) |
| Place of residence | City | 13 (86.7) | 14 (93.3) |
| | Town | 1 (6.7) | 0 (0) |
| | Countryside | 1 (6.7) | 1 (6.7) |
| Average monthly household income (RMB) | <3000 | 0 (0) | 0 (0) |
| | 3000–5000 | 1 (6.7) | 0 (0) |
| | 5000–8000 | 6 (40.0) | 6 (40.0) |
| | >8000 | 8 (53.3) | 9 (60.0) |
| Gestational Age (weeks) | | 30.53 ± 3.38 | 30.67 ± 3.62 |

Note: Mean ± Standard deviation; Numbers (%). BMI body mass index.

**Manipulation check.** The correct response rate for both message detail questions was 100%, indicating that participants carefully reviewed the materials and that the questionnaire was valid. The 100% correct response rate for both message framings in the manipulation check question confirmed that the independent variable was easily identifiable and that the materials were well-designed.

## Results of the pre-survey II

**Characteristics of the participants.** We recruited 60 pregnant women with GDM in strict accordance with the inclusion of exclusion criteria, in which most of the participants were younger than 35 years old (81.8%), had a normal pre-pregnancy BMI (61.7%), were Han Chinese (96.7%), had a bachelor's degree (50%), were settled in an urban area (86.7%), had an average monthly household income of more than 8000 per month (45.0%), and were in the late pregnancy (65.0%) (Table 6).

**Message framing effects test.** The results showed that the messages agreement of the two framings exhibited a significant difference; participants showed more positive attitudes toward loss-framed healthy dietary information compared to gain-framed messages (P < 0.05). Furthermore, pregnant women with GDM were more likely to believe the loss-framed messages on the immediate consequences dimension (P < 0.05). Loss-framed messages about the immediate consequences of fruit and type of food processing on mothers and infants were more convincing (P < 0.05) (Table 7).

## Discussion

The continuously increasing prevalence of GDM has posed a serious threat to the field of public health [2]. Promoting healthy eating behaviors is a key strategy in managing GDM, with the selection of the appropriate dietary information being crucial for improving the level of healthy dietary self-management among pregnant women with GDM [15,16]. However,

**Table 6. Characteristics of the study participants (n = 60).**

| Characteristic | | N | Component Ratio (%) |
|---|---|---|---|
| Age (years) | <35 | 49 | 81.7 |
| | ≥35 | 11 | 18.3 |
| Pre-pregnancy BMI (kg/m²) | < 18.5 | 4 | 6.7 |
| | 18.5-< 24 | 37 | 61.7 |
| | 24-< 28 | 14 | 23.3 |
| | ≥ 28 | 5 | 8.3 |
| Ethnicity | Han nationality | 58 | 96.7 |
| | Minority | 2 | 3.3 |
| Educational level | Junior high school degree and below | 4 | 6.7 |
| | High school or special secondary school degree | 2 | 3.3 |
| | College degree | 14 | 23.3 |
| | Bachelor degree | 30 | 50.0 |
| | Postgraduate degree and above | 10 | 16.7 |
| Place of residence | City | 52 | 86.7 |
| | Town | 5 | 8.3 |
| | Countryside | 3 | 5.0 |
| Average monthly household income (RMB) | <3000 | 3 | 5.0 |
| | 3000–5000 | 11 | 18.3 |
| Gestational Age (weeks) | 5000–8000 | 19 | 31.7 |
| | >8000 | 27 | 45.0 |
| | <28 | 21 | 35.0 |
| | ≥28 | 39 | 65.0 |

**Table 7. Results of the message framing effects test (n = 60).**

| Item | | | M (P25, P75) | | | Statistical value (Z) | |
|---|---|---|---|---|---|---|---|
| Immediate consequences | Staple foods | Gain framing | 5.00 (4.00, 5.00) | Gain framing | 4.57 (4.14, 5.00) | −0.258[a] | −2.570[b],* |
| | | Loss framing | 5.00 (4.00, 5.00) | Loss framing | 5.00 (4.14, 5.00) | | |
| | Meats | Gain framing | 5.00 (4.00, 5.00) | | | −0.484[a] | |
| | | Loss framing | 5.00 (4.00, 5.00) | | | | |
| | Oils | Gain framing | 5.00 (4.00, 5.00) | | | −0.302[b] | |
| | | Loss framing | 5.00 (4.00, 5.00) | | | | |
| | Fruits | Gain framing | 5.00 (4.00, 5.00) | | | −2.530[b],* | |
| | | Loss framing | 5.00 (4.00, 5.00) | | | | |
| | Vegetables | Gain framing | 5.00 (4.00, 5.00) | | | −0.794[b] | |
| | | Loss framing | 5.00 (4.00, 5.00) | | | | |
| | Cooking styles | Gain framing | 5.00 (4.00, 5.00) | | | −1.903[b] | |
| | | Loss framing | 5.00 (4.00, 5.00) | | | | |
| | Types of food processing | Gain framing | 5.00 (4.00, 5.00) | | | −2.507[b],* | |
| | | Loss framing | 5.00 (4.00, 5.00) | | | | |
| Long-term consequences | Staple foods (1) | Gain framing | 5.00 (4.00, 5.00) | Gain framing | 5.00 (4.00, 5.00) | −0.412[a] | −0.399[a] |
| | | Loss framing | 5.00 (4.00, 5.00) | Loss framing | 5.00 (4.00, 5.00) | | |
| | Staple foods (2) | Gain framing | 5.00 (4.00, 5.00) | | | −1.265[a] | |
| | | Loss framing | 5.00 (4.00, 5.00) | | | | |
| | Oils | Gain framing | 5.00 (4.00, 5.00) | | | −0.905[b] | |
| | | Loss framing | 5.00 (4.00, 5.00) | | | | |
| | Meal patterns | Gain framing | 5.00 (4.00, 5.00) | | | −1.000[a] | |
| | | Loss framing | 5.00 (4.00, 5.00) | | | | |
| All | | Gain framing | 4.64 (4.09, 5.00) | | | −2.388[b],* | |
| | | Loss framing | 5.00 (4.09, 5.00) | | | | |

Note:

[a] Gain framing > Loss framing;

[b] Loss framing > Gain framing;

*$P < 0.05$; **$P < 0.01$; ***$P < 0.001$.

effectively guiding the decision-making process and enhancing the reception of dietary information in this population remains an urgent challenge. The message framing effect has demonstrated a significant superiority in the health field [21,22], yet no study has specifically focused on the GDM population. Therefore, this study not only develops a new research tool to promote healthy dietary behaviors in pregnant women with GDM, but also is a preliminary exploration of the message framing effect in this population. The findings will provide a basis for future related studies.

Based on the literature review, the research team collected clinical educational materials on healthy eating and conducted fieldwork at the "one-day ward" and "diabetes outpatient clinic" of the hospital to investigate patients' compliance, information sources, and difficulties. This fieldwork significantly contributed to the development of the study materials. This study found that women with GDM were generally insensitive to food measurements in grams (e.g., "g") and calories (e.g., "kcal"), which is consistent with findings by Yang et al. [38]. As a result, the recommended food quantities to the "palm rule" [39]. During the pre-surveys, participants provided positive feedback, with most of the pregnant women expressing interest in this method, finding it easier to control food portions, and being more willing to implement it. In this study, based on the study of Gao [23], we designed the healthy eating message framing

materials for GDM, addressing different dimensions and topics to help them learn about healthy eating more comprehensively and systematically, while also assessing the effect of the message framings. In addition, we added food pictures to increase patients' reading interest and improve the experimental effect [40]. To eliminate the influence of confounding variables, the number of words and word frequency of the stimulus materials of different message framings were minimized to ensure that the participants faced similar levels of comprehension difficulty.

In the preliminary applied research of this study, the existence of message framing effects in decision-making about healthy eating information for pregnant women with GDM was confirmed and patients were found to prefer loss-framed messages. The possible explanation is that the materials developed in this study were easily recognized as risky information by the participants, and when the riskiness of the information is perceived, individuals are more willing to believe and accept loss-framed messages [41]. In addition, most of the participants in this study were in the late stages of pregnancy and were more likely to fall into a negative emotional state due to disease concerns and limited movement due to larger gestational weeks, potentially suboptimal glycemic control, and fear of adverse pregnancy outcomes [42,43]; if individuals are in a negative mood, the in order to eliminate current negative emotions, they are more likely to favor risk-seeking and adventurous behaviors [44]. Therefore, pregnant women with GDM in this study were more likely to invest more cognitive resources in loss-framed messages and show positive attitudes toward loss-framed messages. There is an extreme paucity of research on message framing effects in GDM populations, and more research is urgently needed to further explore and validate.

Additionally, this study found that the loss-framed messages were more compelling in conveying the immediate consequences of fruit and type of food processing on mothers and infants, while no statistical significance was observed for other food topics. This finding is consistent with Budding et al.'s study [45], which found that loss-framed messages effectively encouraged fruit consumption among adults. However, studies have suggested that loss-framed messages are more likely to decrease participants' willingness to purchase processed and red meats [46,47], whereas no significant difference was found for the meat theme in this study. One possible explanation for this discrepancy is that this study involved a small sample size, which may have limited the findings. Furthermore, fewer studies have examined the message framing effects in relation to specific food topics. Therefore, more research is urgently needed to expand the sample size for further in-depth, targeted exploration and validation of these findings.

In this study, the proportion of female experts was higher, which reflects the predominance of women in fields such as obstetrics, gynecology, and nutrition [48]. However, this gender imbalance may affect the representativeness of the expert panel and introduce potential bias. Therefore, future studies should aim to achieve a more balanced representation to minimize such biases and enhance the diversity of expert perspectives. Furthermore, the generalizability of these findings may be limited due to the study being conducted at a tertiary hospital in Nanjing, Jiangsu Province, with a sample population primarily composed of urban, Han Chinese pregnant women. Urban residents generally have higher education levels, better health awareness, and greater access to health information [49,50]. Moreover, the dietary habits of ethnic minorities differ significantly from those of the Han Chinese [51], which could also influence the study's outcomes. Thus, when generalizing these findings to a broader and more diverse population, factors such as culture, geography, and socioeconomic status should be considered. This study is the first exploration of message framing effects in a GDM population and breaks with the

limitations of previous studies, which have been restricted to perspectives centered solely on the quality of information content. We used information presentation as an entry point to develop targeted and systematic message framing materials specifically for GDM. These materials for this study were evaluated using the Delphi survey method, ensuring their scientific rigor and authority. The manipulation of the independent variables was checked through the pre-survey I, achieving a 100% success rate, indicates that the experimental materials were well-designed. The preliminary use of the compiled materials confirmed the existence of the message framing effects in healthy dietary decision-making among pregnant women with GDM, providing strong support for future studies on the role of message framing in promoting healthy eating behaviors for this population.

For future research in the field of GDM, this study makes the following recommendations:

1. Combining images or videos in the information may be more effective than pure text information [40].

2. Exploring the intrinsic mechanisms by combining advanced experimental tools, such as the more commonly used techniques that can detect neuronal activity characteristics: fMRI (functional magnetic resonance imaging) [52,53], EEG/ERPs (Electroencephalogram/Event Rated Potentials) [54]; and an experimental method commonly used in decision-making scenarios: Eye-movement tracking [55].

3. Currently, most studies in this area are cross-sectional, and it remains uncertain whether the experimental results accurately represent real behavioral changes. Therefore, this study recommends that future research incorporate randomized controlled trials (RCTs) to validate these findings and increase their clinical applicability and generalizability. Specific recommendations for future studies include:

   - Randomized Controlled Trials (RCTs): Randomly assign GDM patients to the gain-framed, loss-framed, and routine groups, receiving targeted information on various food topics. Additional tools, such as food frequency questionnaires and health metrics, should be integrated. The study could be guided by theoretical frameworks such as the Trans-Theoretical Model (TTM) and the Health Belief Model (HBM), continuously and dynamically monitor patient adherence and behavioral changes.

   - Integration into Existing GDM Education Programs: Based on the study's findings, the message framing materials can be integrated into GDM education programs, particularly during the process of healthy dietary management and behavior change. This could involve personalized education interventions, interactive courses (both online and offline), and continuous long-term support.

   - Cross-Disciplinary Collaboration and Broader Dissemination: The research team can collaborate with experts from multiple disciplines to promote the application of message framing materials. The dissemination can be expanded through community health centers, online health platforms, and other public health channels, thus ensuring the broader reach and impact of the intervention materials

4. Combining the mediator and moderator variables with the comprehensive analyses, such as socio-economic status, need for cognition, self-efficacy, risk perception, with comprehensive analyses to explore and explain their potential influence and mechanisms in the effects of message framing on behaviors and intentions [56].

5. Developing a measurable instrument for the message framing effects will facilitate its efficient use in promoting health behaviors.

## Limitations

There are several limitations to this study. Firstly, the materials developed in this study were only pre-surveyed, and whether the stimulus materials have a supportive effect on the attitudes, intentions, or behaviors of pregnant women with GDM requires further exploration and investigation. Secondly, only stimulus materials were provided in this study, and the specific implementation protocols need to be set by future researchers according to the purpose of the studies. Thirdly, the materials of this study put together information on pregnant mothers and their babies, but it is recommended to consider the effects of framing information from both perspectives of the pregnant mother's own risk and the baby's risk on the experimental results in future studies. Lastly, the small sample size selected in this study may affect the results and their generalizability. Therefore, future research should increase the sample size and enhance its representativeness to improve the broader applicability of the findings.

## Conclusions

This study used the Delphi survey method to develop scientifically valid message framing materials for healthy diets in gestational diabetes, and preliminary applications confirmed the existence of message framing effects. Directions and recommendations are provided for future research to facilitate exploration and study of the effects of message frames in this area. This study also provided new perspectives and strategies for effective dissemination of healthy eating education information. It is helpful for the implementation of future related research.

## Supporting information

**S1 Appendix. Items in message framing materials.**
(PDF)

## Acknowledgments

We are grateful to the experts who participated in the Delphi survey.

## Author contributions

**Conceptualization:** Xia Han, Nuo Xu, Jingjing Zhang.

**Data curation:** Xia Han.

**Formal analysis:** Nuo Xu.

**Funding acquisition:** Jingjing Zhang, Ping Gu.

**Investigation:** Xia Han, Nuo Xu, Sijing Chen.

**Methodology:** Jingjing Zhang, Ping Gu.

**Project administration:** Xia Han.

**Software:** Xia Han, Nuo Xu.

**Supervision:** Jingjing Zhang, Ping Gu.

**Validation:** Sijing Chen, Jingjing Zhang, Ping Gu.

**Visualization:** Sijing Chen.

**Writing – original draft:** Xia Han, Nuo Xu.

**Writing – review & editing:** Sijing Chen, Jingjing Zhang, Ping Gu.

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
