## [Decision Letter · Decision Letter 0]

4 Dec 2024

PONE-D-24-46448Message framing materials applied to healthy eating decision-making for pregnant women with gestational diabetes mellitus: An Exploratory StudyPLOS ONE

Dear Dr. gu,

Thank you for submitting your manuscript to PLOS ONE. After careful consideration, we feel that it has merit but does not fully meet PLOS ONE’s publication criteria as it currently stands. Therefore, we invite you to submit a revised version of the manuscript that addresses the points raised during the review process.

**ACADEMIC EDITOR: ** The manuscript offers valuable insights and is well-structured overall. However, I have a few concerns that need to be addressed to strengthen the paper further: Be sure to:

Statistical Analysis: Please provide a clear and robust rationale for the chosen statistical analysis method. Justifying why this approach was selected will enhance the manuscript's methodological transparency and strengthen the validity of the conclusions.Sample Size: Including a detailed explanation of how the sample size for the pre-surveys was determined would be beneficial. In Table 1, please consider including the percentage of male and female participants on the expert panel. If there is a dominant representation of one gender, please justify this or discuss its limitations. Providing this demographic information is crucial for assessing the panel's representativeness and potential biasesClarity and Readability: Review the manuscript for sentence clarity and readability. For example, the sentence on page 4, '*First, participants were informed that they would be participating in an academic study on healthy eating for people with gestational diabetes, and after informed consent was obtained from the participants, they were asked to randomly select a questionnaire, read the corresponding messages carefully, and then completed the questionnaire* ,' is quite long. Consider breaking it into shorter sentences to improve readability and ensure that key information is easily understood. 

A rebuttal letter that responds to each point raised by the academic editor and reviewers. You should upload this letter as a separate file labeled 'Response to Reviewers'.A marked-up copy of your manuscript that highlights changes made to the original version. You should upload this as a separate file labeled 'Revised Manuscript with Track Changes'.An unmarked version of your revised paper without tracked changes. You should upload this as a separate file labeled 'Manuscript'.

We look forward to receiving your revised manuscript.

Kind regards,

Anh Nguyen

Academic Editor

PLOS ONE

Journal Requirements:

2. Please provide additional details regarding expert's consent. In the ethics statement in the Methods and online submission information, please ensure that you have specified (1) whether consent was informed and (2) what type you obtained (for instance, written or verbal, and if verbal, how it was documented and witnessed).

4. Thank you for stating the following financial disclosure: [This work was supported by the Nanjing Health Science and Technology Development Special Fund Project (YKK2214) and the Priority Academic Program Development of Jiangsu Higher Education Institutions (2018, No. 87).]. Please state what role the funders took in the study. If the funders had no role, please state: "The funders had no role in study design, data collection and analysis, decision to publish, or preparation of the manuscript." If this statement is not correct you must amend it as needed. Please include this amended Role of Funder statement in your cover letter; we will change the online submission form on your behalf.

5. Please ensure that you include a title page within your main document. You should list all authors and all affiliations as per our author instructions and clearly indicate the corresponding author.

6. Please include captions for your Supporting Information files at the end of your manuscript, and update any in-text citations to match accordingly. Please see our Supporting Information guidelines for more information: http://journals.plos.org/plosone/s/supporting-information .

Reviewers' comments:

Reviewer's Responses to Questions

**Comments to the Author**

1. Is the manuscript technically sound, and do the data support the conclusions?

Reviewer #1: Yes

Reviewer #2: Yes

2. Has the statistical analysis been performed appropriately and rigorously? 

Reviewer #1: No

Reviewer #2: Yes

3. Have the authors made all data underlying the findings in their manuscript fully available?

Reviewer #1: Yes

Reviewer #2: Yes

4. Is the manuscript presented in an intelligible fashion and written in standard English?

Reviewer #1: No

Reviewer #2: No

5. Review Comments to the Author

Reviewer #1: Strengths:

The study addresses a significant public health issue by exploring message framing effects in dietary decision-making among pregnant women with gestational diabetes mellitus (GDM), an understudied population.

The Delphi method used to develop the framing materials adds robustness and ensures expert consensus.

The two-phase pre-survey methodology offers a systematic approach to test and refine the materials.

The manuscript contributes novel findings, particularly regarding the preference for loss-framed information in the target population.

Concerns and Recommendations:

Clarity of Study Context:

The manuscript does not clearly specify the study setting or location. It is important to include details about where the study was conducted (e.g., the city, region, or country) to contextualize the findings and enhance their applicability to other settings. This information could be added to the Methods section.

Clarity of Presentation:

Some sections, particularly the methodology and statistical analysis, are densely written. Simplify these parts to ensure they are accessible to a broader audience.

Ensure consistent terminology throughout, especially when discussing framing effects (e.g., gain-framed and loss-framed messages).

Statistical Reporting:

Provide more details about the statistical methods used, such as the rationale for selecting the Wilcoxon signed-rank sum test and its appropriateness for the data.

Clarify the interpretation of P-values in Table 7, as some differences (e.g., between framings for oils and meats) are not statistically significant.

Generalizability:

Discuss the limitations of the study population's homogeneity (e.g., predominantly Han nationality, urban residents) and its implications for generalizing the findings to more diverse populations.

Ethical Considerations:

Ensure that ethical approval and the informed consent process are explicitly detailed in the manuscript body, rather than only in supplementary materials.

Practical Implications and Future Directions:

Elaborate on how the findings could be applied in practice, such as integrating the materials into GDM education programs or testing their effectiveness in a randomized controlled trial.

Consider investigating additional factors (e.g., socio-economic status, psychological variables) that may moderate the observed framing effects.

Visuals and Examples:

Include visuals, such as a flow diagram of the study design or representative examples of the message framing materials, to aid in understanding and engaging the reader.

Abstract and Language:

Revise the abstract to make it more concise and focused, avoiding redundancy.

Proofread the manuscript for typographical and grammatical errors (e.g., "message identity" instead of "message identity scores").

Conclusion: The study provides valuable insights into dietary decision-making for women with GDM. Addressing the concerns above, particularly clarifying the study setting and improving the presentation, will significantly enhance the manuscript's clarity, impact, and generalizability. I recommend revisions before the manuscript is considered for publication.

Reviewer #2: The study has a well-defined aim of developing scientifically effective message framing materials for GDM populations. By engaging 17 experts to reach a consensus, the credibility and rigor of the materials are strengthened. Additionally, the use of two pre-surveys to test manipulation validity and framing effects enhances the reliability of the findings. However, I have the following comments:

Minor revision

1. Expert Involvement: The involvement of 17 experts is mentioned, but Table 2 refers to "19 questionnaires delivered and received." Please clarify this discrepancy.

2. Sample Size: The study utilized a relatively small sample size. Did the authors use any formula to calculate the required sample size for the study? The method section should also explain how the 17 experts were identified and recruited. It is important to acknowledge the need for larger sample sizes in future studies to enhance the generalizability of findings.

3. Presentation Refinement:

o Revise phrases such as “complete in the material detail questions” to “complete detailed questions about the materials.”

o Consider restructuring the "Methods" section for improved clarity and readability. Ensure that each step of the study is outlined distinctly.

6. PLOS authors have the option to publish the peer review history of their article (what does this mean? ). If published, this will include your full peer review and any attached files.

**Do you want your identity to be public for this peer review?** For information about this choice, including consent withdrawal, please see our Privacy Policy .

Reviewer #1: **Yes: ** Olivier Mukuku

Reviewer #2: No

---

## [Author Response · Author response to Decision Letter 1]

31 Dec 2024

Dear Prof. Anh Nguyen and reviewers,

Thank you for your letter and for the reviewers’ comments concerning our manuscript PONE-D-24-46448 titled "Message framing materials applied to healthy eating decision-making for pregnant women with gestational diabetes mellitus: An Exploratory Study". Those comments are all valuable and very helpful for revising and improving our paper, as well as the important guiding significance to our research. We have studied comments carefully and have made correction which we hope meet with approval. Revised portions are marked with different colors in the paper. Below are our point-to-point responses to the valuable comments of Academic editor and Reviewers, including the exact location where the changes can be found in the revised manuscript. We use black bold font for the Reviewers' comments, normal black font for our responses, black italics for the content in the original manuscript, and red italics for the changes in the revised manuscript.

Responds to the comments of academic editor and reviewers:

Academic editor:

Comment 1: Statistical Analysis: Please provide a clear and robust rationale for the chosen statistical analysis method. Justifying why this approach was selected will enhance the manuscript's methodological transparency and strengthen the validity of the conclusions.

Response and Revision: Thank you for your insightful comment. We appreciate your suggestion to clarify the rationale behind our statistical methods. Specifically, we have elaborated on the reasoning for selecting the “Wilcoxon signed-rank sum test” in the analysis of the data from pre-surveyⅡ. The Wilcoxon signed-rank sum test was chosen because the data in pre-surveyⅡ did not follow a normal distribution, by using this method, we were able to ensure the robustness of our statistical analysis while accounting for the distribution characteristics of the data. We hope this additional clarification enhances the transparency and methodological rigor of our study. Thank you again for your valuable feedback.

“A Wilcoxon signed rank sum test was used to examine the message framing effects in healthy eating information decision-making among pregnant women with GDM.”

“ A Wilcoxon signed rank sum test was used to examine the message framing effects in healthy eating information decision-making among pregnant women with GDM. This nonparametric method was chosen because the data in the Pre-Survey II section did not meet the assumption of normality, making parametric tests inappropriate. The Wilcoxon signed-rank sum test is ideal for comparing differences between two related (paired) samples, especially when the data cannot be assumed to follow a normal distribution.[36].”

Reference:

Vetter TR, Mascha EJ. Unadjusted Bivariate Two-Group Comparisons: When Simpler is Better. Anesth Analg. 2018;126: 338. doi:10.1213/ANE.0000000000002636

Comment 2: Sample Size: Including a detailed explanation of how the sample size for the pre-surveys was determined would be beneficial.

Response and Revision: Thank you very much for your professional and constructive comments. We agree that providing a clear and detailed explanation of how the sample size for the pre-surveys was determined would enhance the clarity of the methodology. We have included this detailed explanation in the “Pre-surveys” section to clarify the rationale behind our sample size determination.

Pre-survey I: Based on the sample size calculation method used in previous studies [34], a sample of 30 participants was determined for this pre-survey.

Pre-survey II: G*Power 3.1 software was used to calculate the required sample size. With a medium effect size (d = 0.50), significance level (α = 0.05), and statistical power of 1 - β = 0.90, a minimum of 47 subjects was determined [35]. To account for potential sample ineligibility, the sample size was increased by 15%, resulting in a final required sample size of at least 55 participants. Ultimately, 60 participants took part in the study.

Reference:

Perneger TV, Courvoisier DS, Hudelson PM, Gayet-Ageron A. Sample size for pre-tests of questionnaires. Qual Life Res. 2015;24: 147–151. doi:10.1007/s11136-014-0752-2

Kang H. Sample size determination and power analysis using the G*Power software. J Educ Eval Health P. 2021;18: 17. doi:10.3352/jeehp.2021.18.17

Comment 3: In Table 1, please consider including the percentage of male and female participants on the expert panel. If there is a dominant representation of one gender, please justify this or discuss its limitations. Providing this demographic information is crucial for assessing the panel's representativeness and potential biases.

Response and Revision: Thank you very much for your very professional comments. We do recognize the importance of gender balance for the representativeness of the expert group and to avoid potential bias. In response to your suggestion, we have included the percentage of male and female participants in “Table 1” and provided a brief description of the gender distribution in the “Results” section and acknowledged this in the “Discussion” section, highlighting the potential impact of gender ratio bias on certain opinions, in order to enhance the transparency and objectivity of the study.

“The panel has a higher proportion of female experts.”

“In this study, the proportion of female experts was higher, which reflects the predominance of women in fields such as obstetrics, gynecology, and nutrition [48]. However, this gender imbalance may affect the representativeness of the expert panel and introduce potential bias. Therefore, future studies should aim to achieve a more balanced representation to minimize such biases and enhance the diversity of expert perspectives.”

Reference:

Tan Y, Lv J, Nie X, Wei L, Xu J. Survey on Obstacles to Oral Health Care Services During Pregnancy Among Dentists and Obstetricians in Beijing. J Prev Med Chin PLA. 2020;56–58. doi: 10.13704/j.cnki.jyyx.2020.01.020.

Comment 4: Clarity and Readability: Review the manuscript for sentence clarity and readability. For example, the sentence on page 4, 'First, participants were informed that they would be participating in an academic study on healthy eating for people with gestational diabetes, and after informed consent was obtained from the participants, they were asked to randomly select a questionnaire, read the corresponding messages carefully, and then completed the questionnaire,' is quite long. Consider breaking it into shorter sentences to improve readability and ensure that key information is easily understood.

Response and Revision: Thank you very much for reminding. We tried our best to improve the manuscript and made some improvements to the manuscript. These changes were aimed at enhancing clarity and readability without affecting the content or structure of the paper. And here we did not list the changes but marked in red in the revised paper.

Additionally, we enlisted the assistance of a native English speaker to help refine the language and ensure the manuscript meets the standards of academic writing. We hope the revised manuscript is now more clear and coherent .

Reviewer 1：

Comment 1: The manuscript does not clearly specify the study setting or location. It is important to include details about where the study was conducted (e.g., the city, region, or country) to contextualize the findings and enhance their applicability to other settings. This information could be added to the Methods section.

Response and Revision: Thank you very much for your professional comment on our manuscript. We have refined and provided additional details regarding the study implementation sites in the Methods section.

“This study recruited pregnant women with gestational diabetes mellitus who came to Hospital from February 20 to March 24, 2024, and met the above inclusion criteria. This study was approved by the Medical Ethics Committee of ××× (×××-×××).”

“This study recruited pregnant women with gestational diabetes mellitus who came to hospital in Nanjing, China, between February 20 and March 24, 2024, and met the above inclusion criteria. This study was approved by the Medical Ethics Committee of Women’s Hospital of Nanjing Medical University (2022KY-114)."

Comment 2: Some sections, particularly the methodology and statistical analysis, are densely written. Simplify these parts to ensure they are accessible to a broader audience.

Response and Revision: We sincerely appreciate your insightful suggestions, all of which have been addressed in the revised manuscript. We have streamlined the entire text, with particular focus on simplifying the methodology and statistical analysis sections to enhance both clarity and readability. Given the extent of these revisions, we have not listed them here, but all changes are highlighted in red in the manuscript for your review.

Comment 3: Ensure consistent terminology throughout, especially when discussing framing effects (e.g., gain-framed and loss-framed messages).

Response and Revision: Thank you for pointing this out. We have carefully reviewed the manuscript to ensure consistent terminology throughout, particularly when discussing the framing effects. We have standardized the terms "gain-framed / loss-framed messages”and “gain / loss framing”, and have ensured their consistent use across the manuscript. All revisions have been made accordingly.

Comment 4: Provide more details about the statistical methods used, such as the rationale for selecting the Wilcoxon signed-rank sum test and its appropriateness for the data.

Response and Revision: Thank you for pointing out the importance of clarifying the methodology. We have revised the relevant sections to ensure better clarity and precision. The Wilcoxon signed-rank sum test was chosen because the data in pre-surveyⅡ did not follow a normal distribution, by using this method, we were able to ensure the robustness of our statistical analysis while accounting for the distribution characteristics of the data. We hope this additional clarification enhances the transparency and methodological rigor of our study. Thank you again for your valuable feedback.

“A Wilcoxon signed rank sum test was used to examine the message framing effects in healthy eating information decision-making among pregnant women with GDM.”

“ A Wilcoxon signed rank sum test was used to examine the message framing effects in healthy eating information decision-making among pregnant women with GDM. This nonparametric method was chosen because the data in the Pre-Survey II section did not meet the assumption of normality, making parametric tests inappropriate. The Wilcoxon signed-rank sum test is ideal for comparing differences between two related (paired) samples, especially when the data cannot be assumed to follow a normal distribution.[36].”

Reference:

Vetter TR, Mascha EJ. Unadjusted Bivariate Two-Group Comparisons: When Simpler is Better. Anesth Analg. 2018;126: 338. doi:10.1213/ANE.0000000000002636

Comment 5: Clarify the interpretation of P-values in Table 7, as some differences (e.g., between framings for oils and meats) are not statistically significant.

Response and Revision: Thank you very much for your constructive comment. Recognizing that statistically insignificant does not necessarily mean that it is not meaningful in practice, we have added a discussion on food topics to the “Discussion” section, acknowledging the possible impact of small sample size on the results and providing recommendations accordingly.

“Additionally, this study found that the loss-framed messages were more compelling in conveying the immediate consequences of fruit and type of food processing on mothers and infants, while no statistical significance was observed for other food topics. This finding is consistent with Budding et al.'s study [45], which found that loss-framed messages effectively encouraged fruit consumption among adults. However, studies have suggested that loss-framed messages are more likely to decrease participants' willingness to purchase processed and red meats[46,47], whereas no significant difference was found for the meat theme in this study. One possible explanation for this discrepancy is that this study involved a small sample size, which may have limited the findings. Furthermore, fewer studies have examined the message framing effects in relation to specific food topics. Therefore, more research is urgently needed to expand the sample size for further in-depth, targeted exploration and validation of these findings.”

Reference:

de Bruijn G-J, Budding J. Temporal Consequences, Message Framing, and Consideration of Future Consequences: Persuasion Effects on Adult Fruit Intake Intention and Resolve. J Health Commun. 2016;21: 944–953. doi:10.1080/10810730.2016.1179366

Shan L, Jiao X, Wu L, Shao Y, Xu L. Influence of Framing Effect on Consumers’ Purchase Intention of Artificial Meat-Based on Empirical Analysis of Consumers in Seven Cities. Front Psychol. 2022;13: 911462. doi:10.3389/fpsyg.2022.911462

Caso G, Rizzo G, Migliore G, Vecchio R. Loss framing effect on reducing excessive red and processed meat consumption: Evidence from Italy. Meat Sci. 2023;199: 109135. doi:10.1016/j.meatsci.2023.109135

Comment 6: Discuss the limitations of the study population's homogeneity (e.g., predominantly Han nationality, urban residents) and its implications for generalizing the findings to more diverse populations.

Response and Revision: We sincerely appreciate your thoughtful and insightful comment. When generalizing the findings to broader populations, it is crucial to consider the representativeness of the study sample, including factors such as ethnicity, socioeconomic status, geographic location, and cultural background. As this study's sample was primarily drawn from a tertiary hospital in Nanjing, Jiangsu Province, with most participants being Han Chinese and urban residents, the generalizability of the findings may be limited. We have addressed the limitations related to the study population in the "Discussion" section and provided suggestions for future research aimed at enhancing the external validity of the study.

“Furthermore, the generalizability of these findings may be limited due to the study being conducted at a tertiary hospital in Nanjing, Jiangsu Province, with a sample population primarily composed of urban, Han Chinese pregnant women. Urban residents generally have higher education levels, better health awareness, and greater access to health information [49,50]. Moreover, the dietary habits of ethnic minorities differ significantly from those of the Han Chinese [51], which could also influence the study's outcomes. Thus, when generalizing these findings to a broader and more diverse population, factors such as culture, geography, and socioeconomic status should be considered.”

Reference:

Aljassim N, Ostini R. Health literacy in rural and urban populations: A systematic review. Patient Educ Couns. 2020;103: 2142–2154. doi:10.1016/j.pec.2020.06.007

Chen X, Orom H, Hay JL, Waters EA, Schofield E, Li Y, et al. Differences in Rural and Urban Health Information Access and Use. J Rural Health. 2019;35: 405–417. doi:10.1111/jrh.12335

Cai Z, Xian J, Xu X, Zhang Z, Araujo C, Sharma M, et al. Dietary Behaviours Among Han, Tujia and Miao Primary School Students: A Cross-Sectional Survey in Chongqing, China. Risk Manag Healthc P. 2020;13: 1309–1318. doi:10.2147/RMHP.S249101

Comment 7: Ensure that ethical approval and the informed consent process are explicitly detailed in the manuscript body, rather than only in supplementary materials.

Response and Revision: Thank you for this very insightful comment. We have examined the Methods section to ensure that the ethical considerations and the informed consent process are clearly described in each part of the study, in compliance with the ethical guidelines for manuscript publication.

Delphi Procedures:

“This study was approved by the Medical Ethics Committee of Women’s Hospital of Nanjing Medical University (2022KY-114). Prior to the survey, the basic requirements and precautions were explained to the experts. Wr

---

## [Editor Report · Decision Letter 1]

3 Feb 2025

Message framing materials applied to healthy eating decision-making for pregnant women with gestational diabetes mellitus: An Exploratory Study

PONE-D-24-46448R1

Dear Dr. gu,

We’re pleased to inform you that your manuscript has been judged scientifically suitable for publication and will be formally accepted for publication once it meets all outstanding technical requirements.

Kind regards,

Anh Nguyen

Academic Editor

PLOS ONE

Additional Editor Comments (optional):

Thank you for your thoughtful revisions. You have effectively addressed my concerns as well as all reviewers' comments. The manuscript is now thoroughly refined and reads smoothly. This research holds significant value for maternal health, offering valuable insights that can contribute to improving outcomes. Therefore, we are pleased to offer publication.

---

## [Editor Report · Acceptance letter]

PONE-D-24-46448R1

PLOS ONE

Dear Dr. Gu,

I'm pleased to inform you that your manuscript has been deemed suitable for publication in PLOS ONE. Congratulations! Your manuscript is now being handed over to our production team.

Kind regards,

on behalf of

Dr. Anh Nguyen

Academic Editor

PLOS ONE